# Molecular Characterization and Expression Profiling of Nuclear Receptor Gene Families in Oriental Fruit Fly, *Bactrocera Dorsalis* (Hendel)

**DOI:** 10.3390/insects11020126

**Published:** 2020-02-16

**Authors:** Pei-Jin Yang, Er-Hu Chen, Zhong-Hao Song, Wang He, Shi-Huo Liu, Wei Dou, Jin-Jun Wang

**Affiliations:** 1Key Laboratory of Entomology and Pest Control Engineering, College of Plant Protection, Southwest University, Chongqing 400715, China; 2Academy of Agricultural Sciences, Southwest University, Chongqing 400715, China

**Keywords:** *Bactrocera dorsalis*, transcription factors, spatiotemporal expression, development

## Abstract

The oriental fruit fly (*Bactrocera dorsalis*) is a pest that causes large economic losses in the fruit and vegetable industry, so its control is a major challenge. Nuclear receptors (NRs) are a superfamily of ligand-dependent transcription factors that directly combine with DNA to regulate the expression of downstream target genes. NRs are closely associated with multiple physiological processes such as metabolism, reproduction, and development. Through sequence searches and analysis, we identified 21 *B. dorsalis* NR genes, all of which contained at least one of the two characteristic binding domains. On the basis of the conserved sequences and phylogenetic relationships, we divided the 21 NR genes into seven subfamilies. All members of the NR0 subfamily and *BdHR83*, which belonged to the NR2E group, lacked ligand-binding domains. The *BdDSF* and *BdHR51*, which also belonged to the NR2Egroup, and *BdE78* (which belonged to the NR1E group) all lacked DNA-binding domains. The *BdDSF* and *BdHR83* sequences were incomplete, and were not successfully amplified. Development- and tissue-specific expression profiling demonstrated that the transcript levels of the 19 NR genes varied considerably among eggs, larva, pupae, and adults, as well as among larval and adult male and female tissues. Our results will contribute to a better understanding of NR evolution and expand our knowledge of *B. dorsalis* physiology.

## 1. Introduction

Nuclear receptors (NRs) are a large family of transcriptional regulators. They are involved in a variety of physiological functions, including controlling embryonic development, regulating cell differentiation, and maintaining homeostasis. NRs and cofactors on target genes interact with the response sequence in the form of monomers, homodimers and heterodimers [1]. In general, NRs bind receptor ligands for transcriptional regulation, and receptor ligands are usually composed of small lipid molecules, such as retinoids or steroids. However, the regulatory ligands for some receptors (orphan receptors) are still unknown or may not exist [2].

NRs are evolutionarily conserved proteins that have been divided into seven distinct subfamilies and contain a characteristic modular structure [3]. The A/B domains contain a transcriptional activation function (AF-1) and are highly variable with very little evolutionary conservation. The C or DNA-binding domain (DBD) is the most highly conserved domain. It contains two typical cysteine-rich zinc finger motifs in tandem that span about 80 amino acids and are involved directly in recognizing hormone response elements. The D domain functions as a hinge between the DBD and the ligand-binding domain (LBD). The LBD (also known as E domain) contains a hydrophobic ligand-binding pocket that is specific to each receptor and is the source of the sequence variability within the LBD. In addition, the LBD domain mediates dimerization and the ligand-dependent transcriptional activation function (AF-2) [4]. The C-terminal contains the F domain, which is not present in all NRs and is highly diverse.

A large number of NR gene sets have been characterized from the whole-genome sequence of many animals. For example, 48 NRs were identified in the human genome [5], over 270 NRs have been found in *Caenorhabditis elegans* [6], and 49 and 47 were found in mouse and rat, respectively [7]. Surprisingly, only 21 NRs were found in *Drosophila melanogaster* [8], 19 in *Bombyx mori* [9], 22 in *Apis mellifera* [10], and 21 in *Tribolium castaneum* [11].

In *D. melanogaster*, seven NR genes that are regulated directly by 20E have been identified, namely EcR, E75, E78, FTZ-F1, HR3, HR4, and HR39 [12]. Among them, E75 is the primary response gene of 20E. The others are secondary response genes that are maximally expressed after 20E-induced protein synthesis has begun [13]. The ligand–receptor complex of 20E–EcR–USP induces the expression of the 20E primary response gene. Then, the cascade of 20E secondary response genes, which is induced by the transcription factor encoded by the 20E primary response gene, amplifies the 20E signal by regulating the expression of the secondary response genes (e.g., HR3, HR4, and FTZ-F1), thus regulating the physiological processes of molting, metamorphosis, and reproduction in insects [14]. Precise regulation of 20E requires not only 20E-dependent NRs, but also NRs that inhibit its transcriptional regulation (e.g., HR78, HR38, and seven-up (SVP)). These NRs weaken the activity of the 20E signaling pathway by inhibiting the normal function of EcR (Ecdysone Receptor) /USP (Ultraspiracle) dimers. Ten NR genes in *D. melanogaster* are not dependent on 20E. They play a key role in the physiological processes related mainly to embryo formation, neurodevelopment, metabolism, and detoxification. For example, the nuclear hormone receptor HR96 mediates resistance to foreign compounds [15] and HR78 plays a major role in promoting the expression of genes in the midgut, suggesting it may contribute to nutrient uptake [16].

The oriental fruit fly (*Bactrocera dorsalis*) is a devastating and adaptable insect (Diptera: Tephritidae) with a wide range of food sources that include 450 different kinds of fruits and vegetables. This pest poses a serious threat to the fruit and vegetable industry and can cause very serious economic losses [17]. The aim of this study was to identify and annotate NRs in the *B. dorsalis* genome, analyze their structural domains, and construct an evolutionary tree. We inferred gene expression by quantitative real-time PCR (qRT-PCR) to validate the key genes that control development and explore the regulatory mechanism of NR genes in *B. dorsalis.*

## 2. Materials and Methods

### 2.1. Insect Cultures

The *B. dorsalis* laboratory population was collected from Hainan Province, China, in 2008. The insects were kept at 27.5 ± 0.5 °C with 75 ± 5% humidity and a photoperiod of 14:10 h (L:D). The larval feed was composed mainly of agar, yeast, sugar, wheat flour, and corn flour. The adult feed was composed mainly of yeast, sugar, honey, and water. The specific proportions of each ingredient in these artificial diets were as described previously [18] .

### 2.2. Total RNA Extraction and cDNA Synthesis

Total RNA of each sample was extracted using TRIzol^®^ reagent (Invitrogen, Carlsbad, CA, USA). The quality and concentration of RNA were both measured by NanoVue UV–Vis spectrophotometer (GE Healthcare Bio-Sciences, Uppsala, Sweden), and the integrity of isolated RNA was checked by 1.0% agarose gel electrophoresis. Prior to cDNA synthesis, total RNA was treated with RQ1 RNase-Free DNase (Promega, Madison, WI, USA) to digest genomic DNA. For first strand cDNA synthesis, 500 ng of total RNA was reverse transcribed using the PrimeScript 1st Strand cDNA Synthesis Kit (Takara, Dalian, China) with random hexamers and oligo (dT) primers, according to the manufacturer’s instructions.

### 2.3. Identification and Annotation of NRs

To identify and characterize the NR-related proteins of *B. dorsalis*, we downloaded NR amino acid sequences of *D. melanogaster* and *B. dorsalis* from FlyBase (http://flybase.org/) and GenBank (http://www.ncbi.nlm.nih.gov). The NR sequences were used as queries in tBLASTn searches to identify homologous and related genes in the *B. dorsalis* genome database (i5K; https://i5k.nal.usda.gov/webapp/blast/). All the thresholds were set to default and we only took the top hit result. The specific *Drosophila melanogaster* sequences we used as queries are listed in Appendix A. Before the BLAST searches, we had confirmed that all the NRs in *Drosophila melanogaster* were unique. In the subsequent BLAST seearches, the highest result of hits was only one, and there were no multiple results that met the requirements. Furthermore, we used the best reciprocal hits of BLAST to check the NRs in *B. dorsalis* and no false positives were recorded. This indicated that no NR duplication events existed in *B. dorsalis.*

### 2.4. Phylogenetic Analysis

To understand the relationship of *B. dorsalis* NRs (BdNRs) with those of a model dipteran species and other closely related species, the NR amino acid sequences from *B. dorsalis*, *D. melanogaster*, *Bombyx mori*, *T. castaneum*, and *Ceratitis capitata* downloaded from the NCBI (National Center for Biotechnology Information) database (https://www.ncbi.nlm.nih.gov/), FlyBase, BeetleBase and the Silkworm Genome Database (http://silkworm.genomics.org.cn/) were used to construct a phylogenetic tree using MEGA 7.0 (Kuma et al., 2016). Subsequently, neighbor-joining trees were built by applying the Poisson correction model and pairwise deletion method for gaps. Bootstrap iterations to assess the robustness of the generated trees were set to 1000 repetitions. The sequences of 90 NRs from five insect species for the phylogenetic analysis are listed in Appendix A.

### 2.5. Domain Analysis

The DBDs and LBDs were identified in the NR protein sequences of *B. dorsalis* using SMART (http://smart.embl-heidelberg.de/) and the Conserved Domains Search (https://www.ncbi.nlm.nih.gov/Structure/cdd/wrpsb.cgi). We used the ExPASy Compute pI/MW tool (http://cn.expasy.org/tools/pi_tool.html) to obtain the theoretical pI (isoelectric point) and molecular weight (MW) of the deduced NR proteins. We chose NR0, NR1 and NR2 three subfamilies from *B. dorsalis* and *D. melanogaster* and used DNAMAN 6 (https://www.lynnon.com/downloads.html) for the multiple sequence alignment. All parameters were set to default.

### 2.6. Temporal and Developmental Expression Profiles

Samples were collected at different development stages (eggs; one-, three-, five-, seven- and nine-day-old larvae; one-, three-, five-, seven- and nine-day-old pupae; and one-, three-, five-, seven- and nine-day-old male and female adults) and from different tissues (midgut, fat body, integument, Malpighian tubule, and central nervous system of third instar larva; and midgut, fat body, Malpighian tubule, ovary, and testis of four-day-old adults). Specific primers for quantitative real-time PCRs (qRT-PCRs) were designed using Primer 3 web (http://primer3.ut.ee/) and synthesized by Invitrogen in Shanghai, China. The qRT-PCRs were conducted in a 10-μL reaction mixture that included 5 μL qPCR Master Mix (Promega, Madison, WI), 0.5 μL cDNA templates (about 400 ng/μL), and 0.5 mM each of forward and reverse primers. The PCR cycles were 95 °C for 2 min, then 35 cycles at 95 °C for 15 s, 55 °C for 30 s and 72 °C for 20 s, with a final extension at 72 °C for 7 min. A single PCR product was verified by melting curve analysis from 65 to 95 °C. Three pairs of primers were used in *BdDSF* and *BdHR83*, but the amplification failed. All primers used for reverse transcription PCR were listed in Appendix A. Three to four independent biological replicates and two technical replicates were performed for each qRT-PCR. Each replication contained at least two individuals for the developmental stage pattern. Eggs and specific tissues were collected and dissected from 20 individuals to obtain enough RNA. The relative expression levels were obtained using the ΔC_T_ method with *α-tubulin* and *rps3* as the reference genes. We confirmed the presence of a single PCR product at the anticipated size by single melting curve and analyzed changes in the transcript levels of the NR genes using SPSS 16.0 (SPSS Inc., Chicago, IL, USA). The spatiotemporal expression level differences were determined by one-way analysis of variance (ANOVA), followed by Tukey’s test at a significance level of 0.05.

## 3. Results

### 3.1. NR Identification and Characterization

We identified 21 NR genes in the *B. dorsalis* genome. The gene names, GenBank accession numbers, pI and MW parameters, and corresponding NRs in *D. melanogaster* are listed in Table 1. The *BdHR83* and *BdDSF* sequences were incomplete, which made subsequent quantitative analysis unreliable. The BdNRs were divided into seven subfamilies. The atypical NRs that contained only one conserved domain (DBD or LBD) were independently classified regardless of the degree of evolutionary correlation. Each subfamily was divided into several groups. The DBD homology among members of the same group was 80%–90% and the LBD homology was 40%–60%. Three members of the NR0 subfamily (*BdKNRL*, *BdKNI*, and *BdEGON*) contained only DBDs (Figure 1). The NR1 subfamily contained five members, *BdE75*, *BdE78*, *BdEcR*, *BdHR3*, and *BdHR96.* The NR2 subfamily contained eight members, most of which belonged to the NR2E group. In addition, the rest contained one NR3, one NR4, two NR5 and one NR6 subfamily members (Table 1).

Most of the NR amino acid sequences were 300–900 amino acids in length, had MWs of 35.7–94.8 kDa, and pIs from 5.0–9.2. BdE75 (1302 aa; 144.7 kDa) and BdHR4 (1573 aa; 169.1 kDa) were exceptions with longer sequences and higher MWs. Most of the NRs had DBDs and LBDs in their sequences; the exceptions were the NRs in the NR0 subfamily and *BdHR83*, which only had DBDs, and *BdE78*, *BdDSF*, and *BdHR51*, which only had LBDs.

### 3.2. Phylogenetic and Structural Analysis

The phylogenetic tree constructed using the 19 NRs of *B. dorsalis* (excluding *BdHR83* and *BdDSF*) and the available NRs from *B. mori*, *C. capitata*, *D. melanogaster*, and *T. castaneum* (Figure 2) showed that the NRs did not cluster in species-specific clades, rather they segregated into the different subfamilies. Most of the NRs were highly similar to the *D. melanogaster* NRs (DmNRs), and the same subfamily proteins clustered into the same branches. Each subfamily represented NRs with distinct functions; for example, one cluster contained NRs that were related to growth and development.

### 3.3. Spatiotemporal Expression Patterns

Developmental stage and tissue-specific expression of the *B. dorsalis* NRs determined by qRT-PCR, are displayed as heatmaps (Figure 3, Figure 4 and Figure 5). The 19 BdNR genes were expressed in all stages from eggs to adult. *BdE75*, *BdHR4*, and *BdHR3* were highly expressed in five- and seven-day-old pupae; *BdSVP* and *BdUSP* expression levels were lowest in eggs and three-day-old adult females, respectively; *BdTLL* was highly expressed in eggs and five- to seven-day-old adult females; and the relative expression levels of *BdKNI* and *BdKNRL* were stable from eggs to adult (Figure 3). The raw expression data were listed in Appendix A.

Among the different adult tissues, *BdTLL* and *BdHR3* were highly expressed in the ovary, whereas the base expression of *BdTLL* was very low in the female Malpighian tubule, male midgut, and fat body (Figure 4). In the larval tissues, *BdTLL*, *BdHR38*, and *BdHR51* were highly expressed in the central nervous system (Figure 5). The raw expression data were listed in Appendix A.

### 3.4. Multisequence Alignment

Conserved regions in the NR0, NR1 and NR2 subfamily of *B. dorsalis and D. melanogaster* NR proteins were presented in Figure 6, Figure 7 and Figure 8. The *B. dorsalis* genome contained three members of the NR0 subfamily. The *BdKNRL*, *BdKNI*, and *BdEGON* receptors were similar to the *D. melanogaster* Knirp receptors *DmKNRL*, *DmKNI*, and *DmEGON* and shared a high identity at their 5’ ends (Figure 6), whereas there was very little similarity at their 3’ ends. The *B. dorsalis* genome contained five members of the NR1 subfamily. These BdNRs were similar to DmNRs (Table 1, Figure 1). They phylogenetically segregated into the NR1 clade and displayed few similarities to other NR groups (Figure 7). The *B. dorsalis* genome contained eight members of the NR2 subfamily. These BdNRs were similar to those of *D. melanogaster*; however, the *BdHR83*, *BdHR51*, and *BdDSF* sequences were incomplete, and only *BdHR51* was successfully amplified (Table 1, Figure 1). The remaining five members all had high homology with members of the *D. melanogaster* NR2 subfamily; they shared high identity at the 5’ ends but had little similarity at the 3’ ends (Figure 8).

## 4. Discussion

### 4.1. NR0 Subfamily

All of the NR0 subfamily genes lacked LBDs, which is characteristic of the NR0A group [3]. The *D. melanogaster* KNRL has a 19 amino acid kni-box motif adjacent to the zinc fingers [19] and this motif was fully conserved in *BdKNRL*, *BdKNI* and *BdEGON*. In *D. melanogaster*, *DmKNRL* and *DmEGON* mediate the orchestration of embryogenesis and cell fate [20]. No NR0B members were detected among the BdNRs based upon similarity with the vertebrate NRs SHP and DAX1.

*DmKNI* and *DmKNRL* were expressed in *D. melanogaster* at the blastocyst stage (Nauber et al., 1988) and a lack of *DmKNI* and *DmKNRL* expression led to head morphology and trachea development distortions in the late development stage [21]. However, when one or the other of the two genes underwent a single mutation, the head morphological defects were not observed, which indicated that they may have functional redundancy. *DmEGON* was expressed in four neurons and was briefly expressed in the embryonic gonad [22]. The time expression patterns of the NR0 subfamily of genes showed that *BdEGON* was highly expressed during *B. dorsalis* pupal development, which indicated it may be involved in the formation and differentiation of the adult nervous system.

### 4.2. NR1 Subfamily

One of these BdNRs belonged to the NR1D group and was named *BdE75* because of its shared similarity with the *D. melanogaster* E75 receptor. Insect E75 represses the action of HR3 by binding nitric oxide as a ligand. Because of E75 can accommodate heme, it can bind small signaling molecules, such as carbon monoxide and nitric oxide, via the heme moiety [23]. Recently, microRNA let-7 was reported to target the E75 gene in the 20E signaling pathway in order to control larval-pupal development in *B. dorsalis* [24].

Insect HR3, a homolog of vertebrate vitamin A (retinoid-related orphan receptor), is a molting-regulated transcription factor that plays an important role in regulating the expression of tissue-specific genes involved in *D. melanogaster* molting and metamorphosis, and also is necessary for abdominal nerve cord formation [25]. From the phylogenetic analysis, it is uncertain whether the missing domains are true deletions or just artifacts. The qRT-PCR analysis showed that BdHR3 expression was significantly higher just prior to the larval-pupal molt stage and in five- and seven-day-old pupae than at any of the other stages. *BdHR3* was highly expressed in the larval midgut and adult ovary, so we speculated that it may also be involved in larval digestion and adult sexual maturation.

E78 was the only member of the NR1E group, and its sequence was only 363 aa, which is shorter than those of the other members of NR1 (Table 1). The E78 sequence was short mainly because it lacked a DBD. In larval *D. melanogaster*, *DmE78* expression was related to changes in molting hormone titer, because it was induced directly by a molting hormone and depended on 20E-induced protein synthesis to increase its expression level [26]. E78 also played a crucial role in proper egg production and for the maternal control of early embryogenesis. The functions of BdE78 are unclear but may include some of the functions of E78 in *D. melanogaster*.

Analysis of the *B. dorsalis* genome revealed the presence of an ecdysteroid receptor (*BdEcR*) that belonged to the NR1H group (Table 1). Alignment analysis revealed that the deduced protein sequence shared an 86% identity with EcR-B1 of *D. melanogaster*, which indicated that this gene was highly conserved during dipteran evolution. Phylogenetic analysis revealed that *BdEcR* was orthologous to the EcR proteins of other insect species. The qRT-PCR analysis showed that BdEcR was expressed at all tested developmental stages, and its expression reached obviously high levels just prior to the larval-pupal molt and in five-day-old pupae when compared with its expression at other stages. Moreover, the BdEcR gene was much more strongly expressed in the gut and Malpighian tubule than in the trachea and fat body, which indicated it may be involved in tissue-specific functions during larval development [27].

The *B. dorsalis* genome contained one NR with significant homology to *D. melanogaster HR96*. This receptor was in the NR1J group and was designated *BdHR96*. The homology between the *BdHR96* and *D. melanogaster* HR96 sequences was not high, which indicated that gene differentiation may have occurred during dipteran evolution. From the phylogenetic analysis, it is uncertain whether the missing domains are true deletions or just artifacts. However, the function of *BdHR96* is unclear. According to qPCR, high expression existed in the midgut of the adult and the Malpighian tubule of the larva. This indicated that HR96 may play roles in fat decomposition in the midgut. HR96 also was found to play an important role in the stress response of a variety of biomes [15].

### 4.3. NR2 Subfamily

The *B. dorsalis* genome contained a single member of the NR2A group that was designated *BdHNF4* because of its similarity to *D. melanogaster* HNF4. In *D. melanogaster*, *DmHNF4* plays an important role in maintaining glucose homeostasis [28]. Our qRT-PCR analysis showed that *BdHNF4* was expressed at all tested developmental stages, except the egg stage, and was expressed in the fat body, Malpighian tubule, and midgut of the third instar larvae; however, its expression was not obvious in adults. The function of *BdHNF4* may be somewhat similar to that of HNF4 in *D. melanogaster.*

Ultraspiracle (USP), a NR2B group member, was detected in the *B. dorsalis* genome and was designated *BdUSP*. In *D. melanogaster*, EcR and USP are induced by 20E and form heterodimers that initiate the molting cascade, and USP regulates (mainly adult) disc differentiation and midgut formation [29]. The high expression occurred in the midgut of the male and the ovary of the female, and also existed in the midgut and Malpighian tubule of the larvae (Figure 4 and Figure 5). The function of *BdUSP* seems somewhat similar to that of *USP* in *D. melanogaster.* In addition, some influence on ovarian development may also exist.

The *B. dorsalis* genome contained a single member of the NR2D group. We designated it *BdHR78* (because it was most closely related to *D. melanogaster* HR78) and *C. capitata* HR78, both of which belong to the NR2D group. HR78 was found to be involved in regulating the expression of genes in the ecdysone and Notch signaling pathways [16]. The functions of *BdHR78* have not been reported; however, our qRT-PCR analysis indicated that *BdHNF4* and *BdHR78* were expressed in the midgut of third instar larvae, which indicated that these two genes may have some similar functions (Figure 5).

The *B. dorsalis* genome contained four members of the NR2E group (Table 1). Homologs of *BdHR83* and *BdDSF* could not be verified because their sequences were incomplete. TLL in *Drosophila melanogaster* played an important role in visual, nervous, and digestive system development [30]. *BdTLL* was significantly highly expressed in the embryo stage and in the ovaries of seven- to nine-day-old females, which indicated it may be involved in the sexual maturation of the ovaries, but this needs to be verified (Figs. 3–5). Notably, although *BdHR51* was highly expressed in the larval central nervous system (Figure 5), its function is still unknown; however, it has been reported to regulate fly wing development and reproduction [31].

The *B. dorsalis* genome contained a single member of the NR2F group. It was designed as BdSVP because of its homology to SVP in *D. melanogaster*. *DmSVP* can form an allodimer with USP and compete with EcR–USP for DNA binding sites. SVP plays an important role in neural development and is involved in the establishment of the central nervous system of *D. melanogaster* embryos [32], as well as in the proliferation and differentiation of different neuroblasts in later developmental stages [33]. The specific regulatory of SVP indicated that it may repress the induction of the FMRFamide gene prior to target contact [34].

### 4.4. NR3 Subfamily

The *B. dorsalis* genome contained only one member of the NR3B group (Table 1, Figure 1). It was designated as *BdERR*. Structurally, *BdERR* was similar to the estrogen-related receptors (ERRs) of mammals, which indicated that ERRs have highly conserved structural characteristics. A very recent study suggested that the *DmEcR* and *DmERR* jointly regulated the expression of glucose metabolism-related genes [35], but their specific functions in *B. dorsalis* are not yet clear. Our qRT-PCR analysis showed that *BdERR* was expressed in all development stages, but no obviously high expression was detected (Figure 3).

### 4.5. NR4 Subfamily

The *B. dorsalis* genome contained only one member of the NR4A group (Table 1, Figure 1). It was designated *BdHR38*. HR38 may be a true orphan receptor, because no ligand binding pocket or coactivator-binding site was found in its 3D structure [36]. In *D. melanogaster*, HR38 was highly expressed in the gut and integument of larva, and its transcript levels fluctuated according to their nutritional status [37]. Our qRT-PCR analysis indicated that *BdHR38* was highly expressed in the central nervous system of the third instar larva but not in the cuticle (Figure 5). At present, we are unable to explain this result.

### 4.6. NR5 Subfamily

The *B. dorsalis* genome contained two members of the NR5 subfamily: *BdFTZ-F1*, a member of the NR5A group and *BdHR39*, a member of the NR5B group (Table 1, Figure 1). Phylogenetic analysis indicated that *BdFTZ-F1* was orthologous to the FTZ-F1 proteins of *D. melanogaster* (Figure 2). Acetylation of FTZ-F1 and histone H4K5 was required for the fine-tuning of ecdysone biosynthesis during *Drosophila* metamorphosis [38]. Our qRT-PCR analysis indicated that FTZ-F1 was not highly expressed during pupation, but showed obvious expression in M A9 (Figure 3). This indicated that FTZ-F1 may have a different function in *B. dorsalis* compared to that of *D. melanogaster*.

HR39 was the only member of the NR5B group. It shared a high sequence similarity with FTZ-F1, and both subtypes were expressed in the later stages of *D. melanogaster* growth and development [39]. Our qRT-PCR analysis indicated that it was expressed obviously in the Malpighian tubule and midgut at larval stage. However, the specific function of HR39 in *B.dorsalis* is not clear, nor is it supported by citation.

### 4.7. NR6 Subfamily

The *B. dorsalis* genome contained one member of the NR6A group (Table 1, Figure 1). From the phylogenetic analysis, it is uncertain whether the missing domains are true deletions or just artifacts. HR4 was found to be regulated by ecdysteroids and, thus, may coordinate the molting process [39]. In *D. melanogaster*, HR4 functions as a repressor or an inducer when mediating growth and maturation [40].

## 5. Conclusions

So far, 21 NR genes have been identified in *B. dorsalis*. Nineteen of these genes were successfully amplified; the exceptions were HR83 and DSF. Phylogenetic analysis showed that members of the BdNR family were highly homologous to DmNR family members. The NR0 subfamily members (*BdKNRL*, *BdEGON*, and *BdKNI*) all lacked LBDs, and *BdHR83* in the NR2 subfamily also lacked an LBD. The other members of the NR2 subfamily (*BdHR51* and *BdDSF*) and *BdE78* in the NR1 subfamily all lacked DBDs. The absence of these domains may lead to functional changes, but the effects of this on specific functional regulatory mechanisms are still not fully understood.

NR proteins have ubiquitous essential functions in regulating many aspects of metazoan physiology. The biological functions of some of the NRs have been established and related ligands have been identified [41]. However, the functions of many of the NRs remain elusive. The identification of NR genes in *B. dorsalis* is the first step to gain more insight into the evolutionary and structural aspects of these important transcription factors. Spatiotemporal transcript profiles provide a foundation for identifying the physiological functions of specific NRs.

## Figures and Tables

**Figure 1 insects-11-00126-f001:**
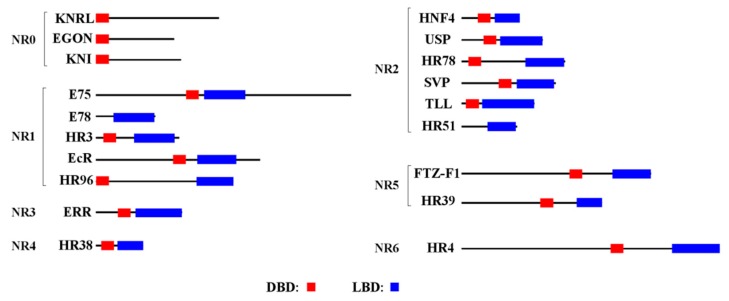
Domain architectures of 21 NR proteins in *B. dorsalis*. DNA-binding domain (DBD) (red); ligand-binding domain (LBD) (blue). The deduced amino acid sequences were used to predict the domain architectures in SMART (http://smart.embl-heidelberg.de/) and NCBI Conserved Domains Search (https://www.ncbi.nlm.nih.gov/Structure/cdd/wrpsb.cgi).

**Figure 2 insects-11-00126-f002:**
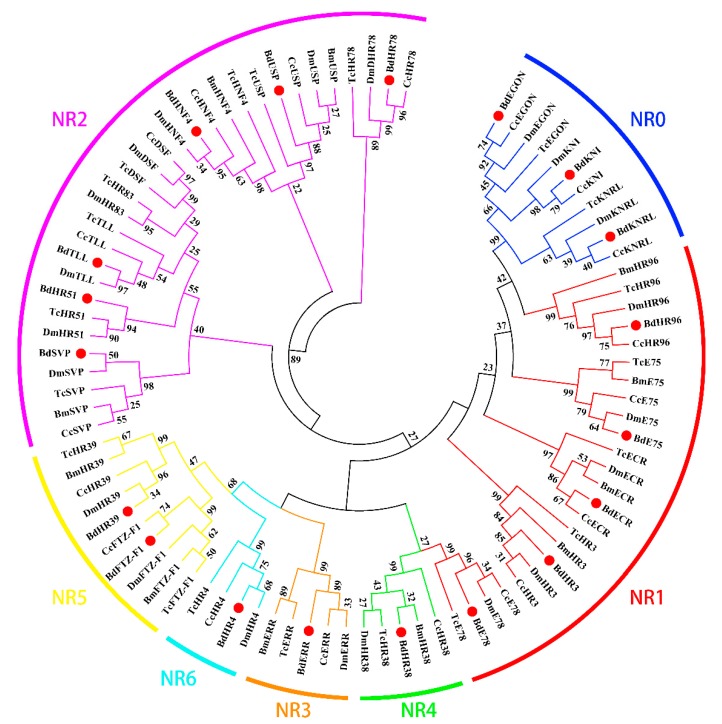
Phylogenetic analysis of 90 insect nuclear receptor (NR) proteins from five different species. *Bombyx mori* (Bm), *Ceratitis capitata* (Cc), *Bactrocera dorsalis* (Bd), *Drosophila melanogaster* (Dm) and *Tribolium castaneum* (Tc). A bootstrap analysis of 1000 replications was carried out on the trees inferred using the neighbor-joining method, and bootstrap values were shown at each branch of the tree. NRs from *B. dorsalis* are indicated by red dots.

**Figure 3 insects-11-00126-f003:**
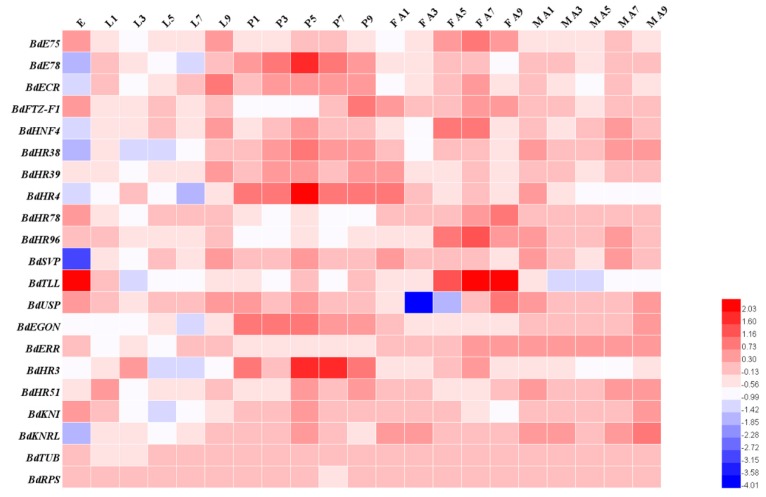
Developmental mRNA expression profiles of 19 *B. dorsalis* NR genes (excluding *BdHR83* and *BdDSF*)**.** Relative transcript levels were calculated using *α-tubulin* (GenBank Accession no.: GU269902) and *rps3* (GenBank Accession no.: XM_011212815) as internal references. Three to four independent biological replicates and two technical replicates were performed for each qRT-PCR. E: eggs; L1, L3, L5, L7, and L9, one- to nine-day-old larva on odd days; P1, P3, P5, P7, and P9, one- to nine-day-old pupae on odd days; F: A1, A3, A5, A7, and A9, one- to nine-day-old female adults on odd days; M A1, A3, A5, A7, A9, one- to nine-day-old male adults on odd days. Warm colors (i.e., red) mean high expression levels and cold colors (i.e., blue) mean low expression levels.

**Figure 4 insects-11-00126-f004:**
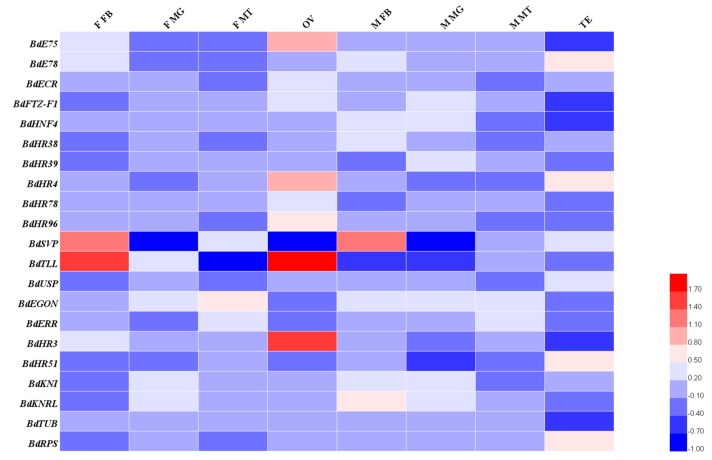
Tissue-specific expression of 19 *B. dorsalis* NR genes (excluding *BdHR83* and *BdDSF*) from four-day-old adults. Relative transcript levels were calculated using *α-tubulin* (GenBank Accession no.: GU269902) and *rps3* (GenBank Accession no.: XM_011212815) as internal references. Three to four independent biological replicates and two technical replicates were performed for each qRT-PCR. Midgut of female/male adults (F/M MG); fat body of female/male adults (F/M FB); Malpighian tubule of female/male adults (F/M MT); ovary of adult females (OV); testis of adult males (TE). Warm colors (i.e., red) indicate high expression levels and cold colors (i.e., blue) indicate low expression levels.

**Figure 5 insects-11-00126-f005:**
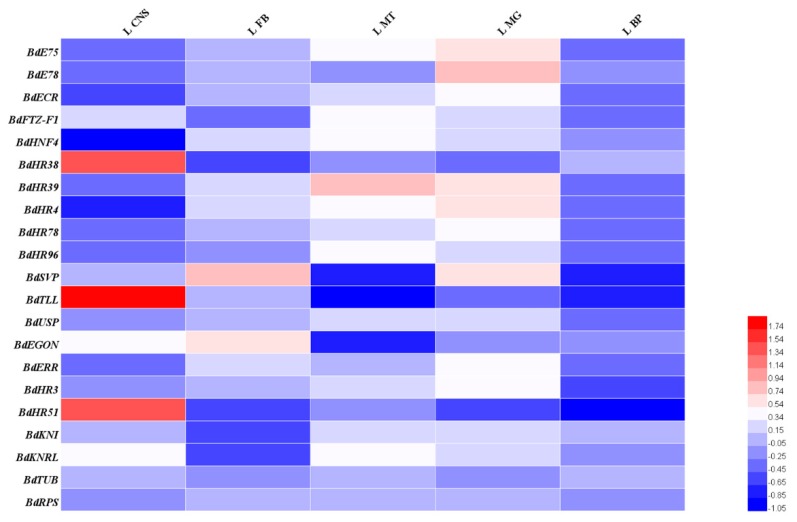
Tissue-specific expression of 19 NR *B. dorsalis* genes (excluding *BdHR83* and *BdDSF*) from third instar larvae. Relative transcript levels were calculated using *α-tubulin* (GenBank Accession no.: GU269902) and *rps3* (GenBank Accession no.: XM_011212815) as internal references. Three to four independent biological replicates and two technical replicates were performed for each qRT-PCR. Larval midgut (L MG); larval fat body (L FB); larval cuticle (L BP); larval Malpighian tubule (L MT); larval central nervous system (L CNS). Warm colors (i.e., red) indicate high expression levels and cold colors (i.e., blue) indicate low expression levels.

**Figure 6 insects-11-00126-f006:**
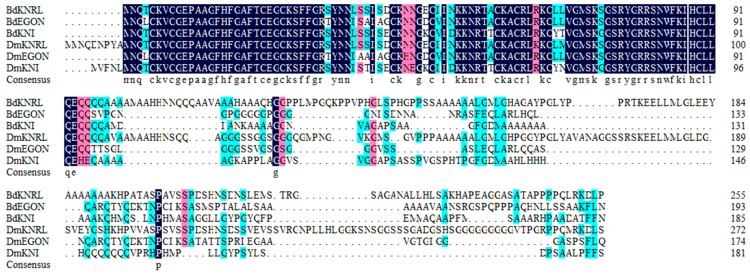
Conserved regions in the NR0 subfamily of *B. dorsalis* and *D. melanogaster* NR proteins. Amino acid sequences of the NR0 subfamily catalytic domains were aligned using DNAMAN 6 (https://www.lynnon.com/downloads.html). Identical and highly conserved amino acids are indicated by black (100%), pink (75%), and blue (50%), respectively.

**Figure 7 insects-11-00126-f007:**
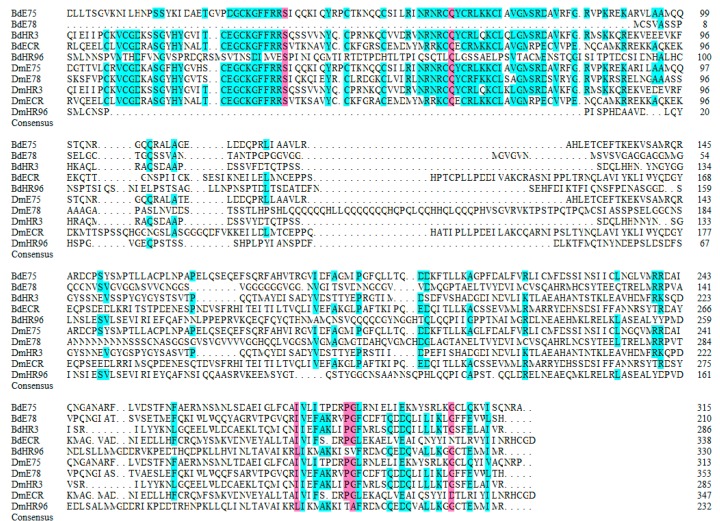
Conserved regions in the NR1 subfamily of *B. dorsalis* and *D. melanogaster* NR proteins. Amino acid sequences of the catalytic domains of the NR1 subfamily were aligned using DNAMAN 6 (https://www.lynnon.com/downloads.html). Identical and highly conserved amino acids are indicated by black (100%), pink (75%), and blue (50%), respectively.

**Figure 8 insects-11-00126-f008:**
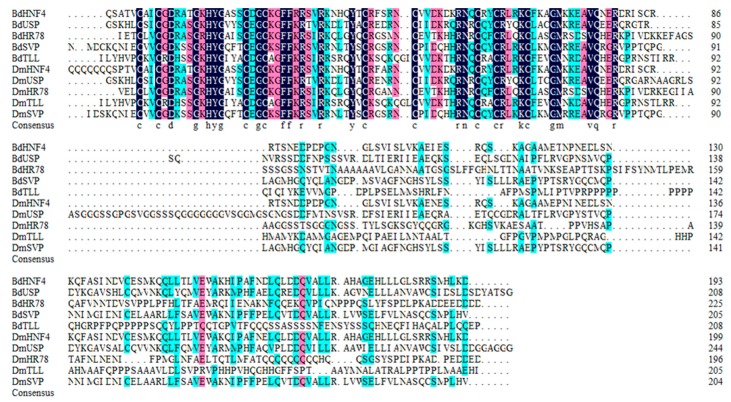
Conserved regions in the NR2 subfamily of NR proteins from *B. dorsalis* (excluding *BdHR51*, *BdDSF*, and *BdHR83*). Amino acid sequences of the catalytic domains of the NR2 subfamily were aligned using DNAMAN 6 (https://www.lynnon.com/downloads.html). Identical and highly conserved amino acids are indicated by black (100%), pink (75%), and blue (50%), respectively.

**Table 1 insects-11-00126-t001:** List of the NR proteins identified in the *B. dorsalis* genome and NRs in *D. melanogaster.*

Group	*B. dorsalis*	Genbank accession no.	Protein length/aa	M.W. /kDa	pI	*D. melanogaster*
0A	BdKNRL	XP_019844421	750	79.4	6.4	KNRL
0A	BdEGON	XP_011213892	477	50.2	9.2	EGON
0A	BdKNI	XP_029404449	519	54.5	6.2	KNI
1D	BdE75	XP_019846264	1302	144.7	8.2	E75
1E	BdE78	XP_019846486	363	38.8	5.2	E78
1F	BdHR3	XP_011208441	485	54.8	6.1	HR3
1H	BdECR	XP_011203444	566	62.8	6.9	ECR
1J	BdHR96	XP_019847193	847	94.8	5.7	HR96
2A	BdHNF4	XP_029404261	698	76.8	6.4	HNF4
2B	BdUSP	XP_019845796	453	51.3	8.5	USP
2D	BdHR78	XP_011201915	630	69.2	5.5	HR78
2E	BdTLL	XP_011214380	516	57.2	5.6	TLL
2E	BdDSF	XP_011214839	603	64.8	7.1	DSF
2E	BdHR83	XP_029407834	318	35.7	9.0	HR83
2E	BdHR51	XP_011200189	339	37.4	5.0	HR51
2F	BdSVP	XP_029407513	574	60.8	8.6	SVP
3B	BdERR	XP_019846473	524	57.4	7.0	ERR
4A	BdHR38	XP_011212592	822	93.1	9.0	HR38
5A	BdFTZ-F1	XP_029405285	751	80.9	6.8	FTZ-F1
5B	BdHR39	XP_019844854	856	89.4	7.3	HR39
6A	BdHR4	XP_019846713	1573	169.1	9.1	HR4

(a) Molecular weight (MW) and isoelectric point (pI) were predicted using the ExPASy ProtParam tool available at http://web.expasy.org/protparam/. (b) Full-length cDNA sequences were not obtained in this study; characteristic parameters were predicted based on the NCBI Reference Sequences.

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
