# Peer review of "Molecular Characterization and Expression Profiling of Nuclear Receptor Gene Families in Oriental Fruit Fly, Bactrocera Dorsalis (Hendel)"

_insects, 2020, doi:10.3390/insects11020126_

Round 1

Reviewer 1 Report

Author summary:

The manuscript Molecular characterization and expression profiling of nuclear receptor gene families in oriental fruit fly, Bactrocera dorsalis (Hendel) by Pei-Jin Yang et. al identifies nuclear receptors (NRs) in the genome of the oriental fruit fly and characterizes their expression pattern. The authors have used Blast to identify all nuclear receptors in the genome and then perform an impressive array of qPCR experiments to show how the expression of these receptors change over time and space.

The paper is well written, presents novel findings, and generally avoids making overly speculative claims. In principle the manuscript deserves to be published. However, there are several changes that the reviewer sees as essential to make sure the results are robust and the conclusions valid.

Major Comments:

Identification of NRs:

In lines 94-98 the authors present their methodology for identifying NRs in B. dorsalis. However, this must be substantially expanded upon. Key questions remain such as

Which type of blast used (Blastp, Blastn, etc.)? Which thresholds were used for expected value (e-value) and query coverage? How many hits that were taken from each query> There are almost always several results that come up from a blast search. Was only the top hit taken? Which specific Drosophila genes were used as queries? Although the names are listed in Table 1, the Flybase gene codes should also be listed in a supplementary table. These are stable over time, while gene names can be variable.

It appears that the authors have found a single 1:1 orthologue for each of the 21 NRs in Drosophila. While this is certainly possible it must be more convincingly shown that there are no NR duplication events in B. dorsalis.

It is also important to note that Blast can be a problematic tool for establishing orthology as it can give false positives due to conserved domains from non-orthologous proteins. Therefore, the reviewer recommends that a more comprehensive search be done. This could include the use of orthology databases like OrthoDB (https://www.orthodb.org/) or by using the best reciprocal hits of BLAST (see Moreno-Hagelsieb et. al 2007 for additional information https://doi.org/10.1093/bioinformatics/btm585).

Completeness of genes

At several points in the manuscript (e.g. Lines 251-252, 279-280, 383-384 among others) the authors discuss several NRs which appear to be missing domains which are found in the Drosophila orthologues. However, it is unclear whether the genes in B. dorsalis actually lack these domains or whether this is an artifact of the gene annotation pipeline used to determine the gene set in this species.

Often times adjacent “genes” on the chromosome are actually exons of the same gene which have been missannotated. Therefore, the reviewer recommends that a more comprehensive manual curation of the genes lacking critical domains be performed by looking at the genomic regions surrounding the truncated genes. Alternatively, the authors could avoid this analysis but must state clearly that it is uncertain whether the missing domains are true deletions or just artifacts.

Add raw expression data

The authors provide a truly impressive amount of qPCR data. While it may not be feasible to fully describe every aspect of the expression values in the current study, many other research groups may find it very useful. The reviewer suggests that the authors include supplementary tables with the quantitative values used to make the heatmap. This way others can more accurately understand the expression data and use it in other applications.

Discussion Length and Content

The discussion is very detailed and well written. However, there are two items which the reviewer believes needs to be addressed prior to publication.

1) Its length is disproportionately large in respect to the rest of the paper. While this would serve as the basis of a good review, the authors should try to restrict the focus of the discussion to only the most relevant points. For example, Lines 300-308 focus almost exclusively on the function of USP in D. melanogaster proteins, and this can be substantially reduced or deleted entirely.

2) Several paragraphs report new data and therefore should be moved up to the results section. Lines 206-209, 230-231, 280-282 describe sequence alignments and produce new figures. These alignments (Figures 6-8) are critical to the paper and provide strong evidence that these genes are truly NRs. Therefore, this material should be moved up to the results section and the methodology behind the alignments should be moved to the methods.

Minor Comments

Lines 16-17: Did the authors use transcriptional analysis to identify NRs? The reviewer thought the expression analysis was only to characterize the NRs identified with Blast.

Lines 27-28 and Lines 392-392: The mention of NRs as insecticide targets must be backed up with more discussion or deleted. At the moment, its inclusion seems overly speculative.

Lines 34-36: Slightly confusing wording. Also, it may be useful to define what a hormone response element is (DNA region?).

Lines 54-59: The citations in this paragraph should be in a similar format to the rest of the paper.

Lines 73: Does HR96 mediate resistance to foreign compounds or organisms? The resistance against compounds is well supported by the citation.

Lines 80-81: It must be emphasized that this sentence is drawn on inferences from model organism like D. melanogaster. Alternatively, the sentence can be deleted.

Lines 101-102: Where were the sequences from B. mori, T. castaneum and C. capitata obtained?

Lines 114-128: The qPCR section is very well done. Please include a supplementary table with the sequences of the NR and reference primers used in this study.

Lines 245-246: It is incorrect to say that BdHR3 was orthologous to EcR protein in other species. BdHR3 clusters with the HR3 protein in other species and this is a sister group to the EcR proteins. The original statement makes it appear that BdHR3 is more closely related to EcR proteins, which is not true.

Lines 285: The reviewer is unfamiliar with a “Markov tube”. Is this a typo for Malpighian tubules?

Suggestions (include at authors discretion):

Paragraph 2 of the introduction (Lines 39-42) appears to interrupt the logical flow of the paper. Maybe remove it.

In figure 4, it might make more sense to group the ovary column with the female measurements and the testis with the male measurements.

Author Response

The manuscript Molecular characterization and expression profiling of nuclear receptor gene families in oriental fruit fly, Bactrocera dorsalis (Hendel) by Pei-Jin Yang et. al identifies nuclear receptors (NRs) in the genome of the oriental fruit fly and characterizes their expression pattern. The authors have used Blast to identify all nuclear receptors in the genome and then perform an impressive array of qPCR experiments to show how the expression of these receptors change over time and space.

The paper is well written, presents novel findings, and generally avoids making overly speculative claims. In principle the manuscript deserves to be published. However, there are several changes that the reviewer sees as essential to make sure the results are robust and the conclusions valid.

Response: Thanks for your positive evaluation.

Major Comments:

Identification of NRs:

In lines 94-98 the authors present their methodology for identifying NRs in B. dorsalis. However, this must be substantially expanded upon. Key questions remain such as Which type of blast used (Blastp, Blastn, etc.)? Which thresholds were used for expected value (e-value) and query coverage? How many hits that were taken from each query> There are almost always several results that come up from a blast search. Was only the top hit taken? Which specific Drosophila genes were used as queries? Although the names are listed in Table 1, the Flybase gene codes should also be listed in a supplementary table. These are stable over time, while gene names can be variable.

Response: We downloaded the amino acid sequences of Drosophila melanogaster from Flybase into tBlastn to identify the NRs in Bactrocera dorsalis. All the default thresholds were used. General Parameters: max target sequences (100), expect threshold (10); Scoring Parameters: matrix (BLOSUM62), gap costs (existence: 11, extension: 1), compositional adjustments (conditional compositional score matrix adjustment). The hits taken from each query were over 75%. Then we only took the top hit result. The specific Drosophila melanogaster genes we used as queries were listed in Table S2.

It appears that the authors have found a single 1:1 orthologue for each of the 21 NRs in Drosophila. While this is certainly possible it must be more convincingly shown that there are no NR duplication events in B. dorsalis.

Response: Before the Blast, we had confirmed that all the NRs in Drosophila melanogaster were unique. In the subsequent blast, the highest result of hits was only one, and there were no multiple results meeting with the requirements. Therefore, we believed that no NR duplication event existed in B. dorsalis.

 It is also important to note that Blast can be a problematic tool for establishing orthologys it can give false positives due to conserved domains from non-orthologous proteins. Therefore, the reviewer recommends that a more comprehensive search be done. This could include the use of orthology databases like OrthoDB (https://www.orthodb.org/) or by using the best reciprocal hits of BLAST (see Moreno-Hagelsieb et. al 2007 for additional information https://doi.org/10.1093/bioinformatics/btm585).

Response: Thank you for your valuable advice. Furthermore, we used the best reciprocal hits of BLAST to check the NRs and no false positives were recorded.

Completeness of genes

At several points in the manuscript (e.g. Lines 251-252, 279-280, 383-384 among others) the authors discuss several NRs which appear to be missing domains which are found in the Drosophila orthologues. However, it is unclear whether the genes in B. dorsalis actually lack these domains or whether this is an artifact of the gene annotation pipeline used to determine the gene set in this species.

Often times adjacent “genes” on the chromosome are actually exons of the same gene which have been missannotated. Therefore, the reviewer recommends that a more comprehensive manual curation of the genes lacking critical domains be performed by looking at the genomic regions surrounding the truncated genes. Alternatively, the authors could avoid this analysis but must state clearly that it is uncertain whether the missing domains are true deletions or just artifacts.

Response: Thank you for your advice. The description “it is uncertain whether the missing domains are true deletions or just artifacts” has been added correspondingly in Discussion section.

Add raw expression data

The authors provide a truly impressive amount of qPCR data. While it may not be feasible to fully describe every aspect of the expression values in the current study, many other research groups may find it very useful. The reviewer suggests that the authors include supplementary tables with the quantitative values used to make the heatmap. This way others can more accurately understand the expression data and use it in other applications.

Response: Thanks for your advice. Raw expression data have been added in Table S4, S5 and S6.

Discussion Length and Content

The discussion is very detailed and well written. However, there are two items which the reviewer believes needs to be addressed prior to publication.

1) Its length is disproportionately large in respect to the rest of the paper. While this would serve as the basis of a good review, the authors should try to restrict the focus of the discussion to only the most relevant points. For example, Lines 300-308 focus almost exclusively on the function of USP in D. melanogaster proteins, and this can be substantially reduced or deleted entirely.

Response: Thank you for your advice. We have adjusted the structure of the discussion and removed the redundant parts to make the paper proportionate.

2) Several paragraphs report new data and therefore should be moved up to the results section. Lines 206-209, 230-231, 280-282 describe sequence alignments and produce new figures. These alignments (Figures 6-8) are critical to the paper and provide strong evidence that these genes are truly NRs. Therefore, this material should be moved up to the results section and the methodology behind the alignments should be moved to the methods.

Response: Thank you for your advice. We have moved the corresponding contents to the results and methods section.

Minor Comments

Lines 16-17: Did the authors use transcriptional analysis to identify NRs? The reviewer thought the expression analysis was only to characterize the NRs identified with Blast.

Response: The mistake was corrected.

Lines 27-28 and Lines 392-392: The mention of NRs as insecticide targets must be backed up with more discussion or deleted. At the moment, its inclusion seems overly speculative.

Response: Thank you for your advice. The overly speculative descriptions have been deleted.

Lines 34-36: Slightly confusing wording. Also, it may be useful to define what a hormone response element is (DNA region?).

Response: Thank you for your advice. “the response sequence” was added to replace “hormone response element”.

Lines 54-59: The citations in this paragraph should be in a similar format to the rest of the paper.

Response: A similar format was corrected for all the citations.

Lines 73: Does HR96 mediate resistance to foreign compounds or organisms? The resistance against compounds is well supported by the citation.

Response: Yes, HR96 mediates resistance to foreign compounds. The mistake was corrected.

Lines 80-81: It must be emphasized that this sentence is drawn on inferences from model organism like D. melanogaster. Alternatively, the sentence can be deleted.

Response: According to your advice, the sentences were deleted.

Lines 101-102: Where were the sequences from B. mori, T. castaneum and C. capitata obtained?

Response: These sequences were downloaded from the NCBI database, FlyBase, BeetleBase and the Silkworm Genome Database. We have listed these sequences and Genbank accession numbers in Table S3.

Lines 114-128: The qPCR section is very well done. Please include a supplementary table with the sequences of the NR and reference primers used in this study.

Response: Thank you for your advice. These data were added and presented in Table S1 and S2.

Lines 245-246: It is incorrect to say that BdHR3 was orthologous to EcR protein in other species. BdHR3 clusters with the HR3 protein in other species and this is a sister group to the EcR proteins. The original statement makes it appear that BdHR3 is more closely related to EcR proteins, which is not true.

Response: Thank you for your advice. The incorrect description was removed.

Lines 285: The reviewer is unfamiliar with a “Markov tube”. Is this a typo for Malpighian tubules?

Response: Yes, it is a typo for “Malpighian tubules”.

Suggestions (include at authors discretion):

Paragraph 2 of the introduction (Lines 39-42) appears to interrupt the logical flow of the paper. Maybe remove it.

Response: Thank you, and we have removed it.

In figure 4, it might make more sense to group the ovary column with the female measurements and the testis with the male measurements.

Response: Thank you, and we have adjusted the figure correspondingly.

Reviewer 2 Report

The manuscript "insects-683764" describes a preliminary characterization of Bactrocera dorsalis nuclear receptors. Using the sequences of the nuclear receptors from Drosophila melanogaster and other insect species as starting point, the authors identified putative B. dorsalis nuclear receptor sequences from an existing, partially annotated genome assembly. Basic phylogenetic analysis showed that the nuclear receptors in this species are as expected, very similar to the ones in Drosophila and other dipterans. The main contribution of the manuscript is the analysis of the expression levels of 19 nuclear receptors at 21 developmental stages, 8 adult tissues and 5 larval tissues by qRT PCR, making a total of 646 samples analyzed. Unfortunately, the authors cannot conclude much from this extensive expression analysis and thus, the interest of the work is very limited. This is reflected in the Discussion section, mostly devoted to review some aspects of the nuclear receptors literature in Drosophila and other species with very few references to the actual results described in the manuscript.

Specific comments:

- Methods section:

 -I find the description of the identification and annotation of nuclear receptors (section 2.2) incomplete. It is not obvious to me if the authors performed any manual annotation of the B. dorsalis nuclear receptor genes or just relied on existing automatic annotations. In particular, the authors mention that the sequences of BdHR83 and BdDSF were incomplete. This could mean that there are gaps in the genome assembly or just that the automatic annotation contains errors that can be manually fixed.

- The authors mention that HR83 and DSF genes "could not be amplified" (line 378). I assume this means that the RT-PCRs did not generate specific products in any of the tissues analyzed. This could indicate that these two genes are expressed at very low levels or that the oligo pairs used are not appropriate. To be able to reach a conclusion from this observation, it should be mentioned in the methods section the number of oligo pairs tested, their sequence and the controls performed to make sure they are able to amplify the desired cDNAs.

- Additiona issues:

- line 35: The authors write "homologous and heterogeneous dimers". It should probably say "homodimers and heterodimers".

-line 36: "transcriptional regulation of NRs relies on the binding of receptor ligands". This sentence appears to indicate that transcription of the NRs depends on their binding to ligands.

-line 39: Molting hormones regulate development, not abnormal development.

-line 63: This section describes NRs regulation by ecdysone in Drosophila, but the reference cited [7] describes work on Daphnia magna.

-line 221. It says: "DmEGON  was  expressed  mainly  in  the  fourth neuron and ...". However, the article cited [17] mentions that DmEGON is expressed in four neurons. This is probably a typo.

-line 233. It says: "HR3 by serving as a dimerization partner to the nitric oxide." NO can bind to HR3 as a ligand. I would not say NO is a "dimerization partner".

-line 284. It says: "In D. melanogaster, DmHNF4 was expressed during liposome development". From my understanding, liposomes are not a part of Drosophila. Not sure what the authors refer to here.

-line 285. It continues: "... and Markov tube and midgut formation". Again, I do not know of any Drosophila organ called Markov tube.

Author Response

The manuscript "insects-683764" describes a preliminary characterization of Bactrocera dorsalis nuclear receptors. Using the sequences of the nuclear receptors from Drosophila melanogaster and other insect species as starting point, the authors identified putative B. dorsalis nuclear receptor sequences from an existing, partially annotated genome assembly. Basic phylogenetic analysis showed that the nuclear receptors in this species are as expected, very similar to the ones in Drosophila and other dipterans. The main contribution of the manuscript is the analysis of the expression levels of 19 nuclear receptors at 21 developmental stages, 8 adult tissues and 5 larval tissues by qRT PCR, making a total of 646 samples analyzed. Unfortunately, the authors cannot conclude much from this extensive expression analysis and thus, the interest of the work is very limited. This is reflected in the Discussion section, mostly devoted to review some aspects of the nuclear receptors literature in Drosophila and other species with very few references to the actual results described in the manuscript.

Response: Thank you for your valuable advice. We have adjusted the structure of the discussion and removed the redundant parts to make the paper proportionate.

Specific comments:

- Methods section:

 -I find the description of the identification and annotation of nuclear receptors (section 2.2) incomplete. It is not obvious to me if the authors performed any manual annotation of the B. dorsalis nuclear receptor genes or just relied on existing automatic annotations. In particular, the authors mention that the sequences of BdHR83 and BdDSF were incomplete. This could mean that there are gaps in the genome assembly or just that the automatic annotation contains errors that can be manually fixed.

Response: Thank you for your advice. The results of this article are based solely on existing automated annotations. We have tried to find the missing pieces of BdHR83 and BdDSF, but it did not work out. Finally, it is uncertain whether the missing domains are true deletions or just artifacts.

- The authors mention that HR83 and DSF genes "could not be amplified" (line 378). I assume this means that the RT-PCRs did not generate specific products in any of the tissues analyzed. This could indicate that these two genes are expressed at very low levels or that the oligo pairs used are not appropriate. To be able to reach a conclusion from this observation, it should be mentioned in the methods section the number of oligo pairs tested, their sequence and the controls performed to make sure they are able to amplify the desired cDNAs.

Response: Thank you for your advice. Three pairs of primers were used in BdDSF and BdHR83 and all primers used for reverse transcription PCR were listed in Table S1.

- Additiona issues:

- line 35: The authors write "homologous and heterogeneous dimers". It should probably say "homodimers and heterodimers".

Response: Thank you for your advice. "homodimers and heterodimers" was corrected to replace "homologous and heterogeneous dimers".

-line 36: "transcriptional regulation of NRs relies on the binding of receptor ligands". This sentence appears to indicate that transcription of the NRs depends on their binding to ligands.

Response: Thank you for your advice. The sentence was changed for specific description.

-line 39: Molting hormones regulate development, not abnormal development.

Response: Thank you for your advice. We have deleted this part for logical continuity.

-line 63: This section describes NRs regulation by ecdysone in Drosophila, but the reference cited [7] describes work on Daphnia magna.

Response: Thank you for your advice. This is an error in our citation and it has been corrected.

The role of nuclear receptor E75 in regulating the molt cycle of Daphnia magna and consequences of its disruption

-line 221. It says: "DmEGON  was  expressed  mainly  in  the  fourth neuron and ...". However, the article cited [17] mentions that DmEGON is expressed in four neurons. This is probably a typo.

Response: Thank you for your advice. We have corrected it.

-line 233. It says: "HR3 by serving as a dimerization partner to the nitric oxide." NO can bind to HR3 as a ligand. I would not say NO is a "dimerization partner".

Response: Thank you for your advice. For specific description, the sentence was changed to “Insect E75 represses the action of HR3 by binding nitric oxide as a ligand”.

-line 284. It says: "In D. melanogaster, DmHNF4 was expressed during liposome development". From my understanding, liposomes are not a part of Drosophila. Not sure what the authors refer to here.

Response: Thank you for your advice. It is a typo in our statement and it indeed refers to the liposome of the larva.

-line 285. It continues: "... and Markov tube and midgut formation". Again, I do not know of any Drosophila organ called Markov tube.

Response: Thank you for your advice. It is a typo and we have corrected it into “Malpighian tubule”.

Reviewer 3 Report

This study aims to annotate the Nuclear Receptor (NR) genes in the oriental fruit fly, Bactrocera dorsalis. Both bioinformatic and experimental methodologies are utilised, yielding insights into the evolutionary relatedness, as well as the temperospatial expression dynamics of this gene superfamily in B. dorsalis. The scientific merit of this work is justified, although a number of improvements (described below) would increase the overall impact.

Analysis of the gene expression data (qRT-PCR).

It would be appropriate to present these data in the form of boxplots with the inclusion of relevant descriptive statistics. The presentation as it currently stands (i.e. heatmaps) is difficult to interpret as the quantitative values (i.e. colour between red and blue) are not clearly explained. In addition, there is no mention of statistical significance (i.e. P values) in the results section and it is unclear if biological replicates were included despite this being mentioned in the methods section.

What was the level of biological replication for each developmental stage/tissue type and was this replication consistent across all the genes analysed? I suggest this information be included in the appropriate figure legends and statistically significant differences highlighted.

Other comments.

It would be useful to include the Genbank accession numbers for sequences used in the various analyses (i.e. BLAST searches, phylogenetic analysis and multiple sequence alignments).

It would also be worthwhile to include the sequences of all PCR primers used throughout the study and to confirm the presence of a single PCR product at the anticipated size.

Author Response

This study aims to annotate the Nuclear Receptor (NR) genes in the oriental fruit fly, Bactrocera dorsalis. Both bioinformatic and experimental methodologies are utilised, yielding insights into the evolutionary relatedness, as well as the temperospatial expression dynamics of this gene superfamily in B. dorsalis. The scientific merit of this work is justified, although a number of improvements (described below) would increase the overall impact.

Analysis of the gene expression data (qRT-PCR).

It would be appropriate to present these data in the form of boxplots with the inclusion of relevant descriptive statistics. The presentation as it currently stands (i.e. heatmaps) is difficult to interpret as the quantitative values (i.e. colour between red and blue) are not clearly explained. In addition, there is no mention of statistical significance (i.e. P values) in the results section and it is unclear if biological replicates were included despite this being mentioned in the methods section.

Response: Thank you for your advice. For the inclusion of relevant descriptive statistics, Tables S4, S5 and S6 were added to present the significance and raw expression data. Three to four independent biological replicates and two technical replicates were performed for each qRT-PCR.

What was the level of biological replication for each developmental stage/tissue type and was this replication consistent across all the genes analysed? I suggest this information be included in the appropriate figure legends and statistically significant differences highlighted.

Response: Thank you for your advice. Three to four independent biological replicates and two technical replicates were performed for each qRT-PCR. This replication was consistent across all the genes analyzed. Besides, we provided the significance and raw expression data in Table S4, S5 and S6.

Other comments.

It would be useful to include the Genbank accession numbers for sequences used in the various analyses (i.e. BLAST searches, phylogenetic analysis and multiple sequence alignments).

Response: Thank you for your advice. We have listed these data in Table S2 and S3.

It would also be worthwhile to include the sequences of all PCR primers used throughout the study and to confirm the presence of a single PCR product at the anticipated size.

Response: Thank you for your advice. We have listed all PCR primers used throughout the study in Table S1 and confirmed the presence of a single PCR product at the anticipated size by single melting curve.

Reviewer 4 Report

In this study, Yang et al. used online sources of amino acid sequences to identify nuclear receptor (NR) genes in the online-published Bactrocera dorsalis genome. Next, the expression profiles of these NR genes were characterized in different B. dorsalis tissue (midgut, fat body, integument, Malpighian tubule, central nervous system, ovary and testis) and during different developmental phases (larvae, pupae and adults). Interestingly, previously identified functions of some of these NR genes are in accordance with the tissue and timing of their expression levels presented here.

In summary, the article is very well written and the presented results are an excellent example of using the high amount of gathered resources from Drosophila melanogaster to study in B. dorsalis. Furthermore, the authors present their results in a broad dipteran evolutionary fashion by comparing NR gene sequences from several dipteran species, albeit using data from online sources.

Major concerns:

>RNA extraction method is not included in the methods section. Was it easy to obtain enough RNA from the specific tissue of single individuals or were they pooled?

>I think it would be good to include the exact number of individuals used in this study. Also, the raw expression data should be provided as supplementary information. Furthermore, the statistical significance testing of expression level differences is stated but these results are not shown anywhere in the paper. For example, in L262 it states that BdEcR reached significantly high expression levels... . Does that mean that the other reports of high (or low) expression levels are not significant?

Also, in figures 3, 4 and 5, does an expression value of 2 mean 2-fold more expressed than the control gene? And if so, which of the two control genes (rps3, alpha-tubulin). This could be clearer.

>Figures 7 & 8 as well as homology between species should be presented as results instead of in discussions. Also, legend should state clearly that these figures compare B. dorsalis sequences with D. melanogaster sequences (at the moment the legend is not very clear).

>In discussions, some paragraphs discuss known functions of tested NR genes but no relation to the actual results is presented. For example, paragraph beginning on L300. What was found in this study and how does this information relate to that? Same with paragraphs starting on lines 330, 365 and 372.

Minor concerns:

>some references are cited by name and year in the text, whereas others by chronological numbering. This should be standardised.

>L118: quantitative real-time PCR or qRT-PCR. I think “quantitative real-time RT-PCR” is a redundant use of the abbreviation.

>L125: mis-spelling of “rps3as”.

>L135: Mis-spelling of “CBD”. I believe the authors wished to write “DBD”.

>L251: “The E78 sequence was short mainly because it lacked a DBD.” How does this finding compare to other insect species compared in this study? i.e., Is this specific to B. dorsalis?

>L233: type-o in “Because of E75 can accomodate heme...”

>In discussions, some of the speculative conclusions could be followed by how to definitely answer the functions of these NRs in B. dorsalis.

>L310: Type-o closed parenthesis: “... related to D. melanogaster HR78) ...”

Author Response

In this study, Yang et al. used online sources of amino acid sequences to identify nuclear receptor (NR) genes in the online-published Bactrocera dorsalis genome. Next, the expression profiles of these NR genes were characterized in different B. dorsalis tissue (midgut, fat body, integument, Malpighian tubule, central nervous system, ovary and testis) and during different developmental phases (larvae, pupae and adults). Interestingly, previously identified functions of some of these NR genes are in accordance with the tissue and timing of their expression levels presented here.

In summary, the article is very well written and the presented results are an excellent example of using the high amount of gathered resources from Drosophila melanogaster to study in B. dorsalis. Furthermore, the authors present their results in a broad dipteran evolutionary fashion by comparing NR gene sequences from several dipteran species, albeit using data from online sources.

Response: Thanks for your positive evaluation.

Major concerns:

>RNA extraction method is not included in the methods section. Was it easy to obtain enough RNA from the specific tissue of single individuals or were they pooled?

Response: Thank you for your advice. We have added this section on 2.2. We used 2 individuals of biological replication for each developmental stage. Eggs and specific tissues were collected and dissected from 20 individuals to obtain enough RNA.

>I think it would be good to include the exact number of individuals used in this study. Also, the raw expression data should be provided as supplementary information. Furthermore, the statistical significance testing of expression level differences is stated but these results are not shown anywhere in the paper. For example, in L262 it states that BdEcR reached significantly high expression levels... . Does that mean that the other reports of high (or low) expression levels are not significant?

Response: Thank you for your advice. We added the exact number of individuals used in this study at methods section. The raw expression data were listed in Table S4, S5 and S6. By checking the significance testing of expression, we have revised our wording to “obviously” but not “significantly”.

Also, in figures 3, 4 and 5, does an expression value of 2 mean 2-fold more expressed than the control gene? And if so, which of the two control genes (rps3alpha-tubulin). This could be clearer.

Response: Thank you for your advice. The heatmaps data used log10 transformation. So the expression value of 2 mean 102 (100-fold) more expressed than the control gene. The alpha-tubulin was used to generate the heatmaps.

>Figures 7 & 8 as well as homology between species should be presented as results instead of in discussions. Also, legend should state clearly that these figures compare B. dorsalissequences with D. melanogaster sequences (at the moment the legend is not very clear).

Response: Thank you for your advice. We have moved them to the results section and made it clear that these figures compare Bactrocera dorsalis sequences with Drosophila melanogaster sequences.

>In discussions, some paragraphs discuss known functions of tested NR genes but no relation to the actual results is presented. For example, paragraph beginning on L300. What was found in this study and how does this information relate to that? Same with paragraphs starting on lines 330, 365 and 372.

Response: Thank you for your advice. We have adjusted the structure of the discussion and removed the redundant parts to make the paper proportionate.

Minor concerns:

>some references are cited by name and year in the text, whereas others by chronological numbering. This should be standardised.

Response: Thank you for your advice. We have checked the references and standardized the format.

>L118: quantitative real-time PCR or qRT-PCR. I think “quantitative real-time RT-PCR” is a redundant use of the abbreviation.

Response: Thank you for your advice. We have deleted “RT-”.

>L125: miss-pelling of “rps3as”.

Response: Thank you and we have changed into “rps3”.

>L135: Mis-spelling of “CBD”. I believe the authors wished to write “DBD”.

Response: Thank you and we have changed into “DBD”.

>L251: “The E78 sequence was short mainly because it lacked a DBD.” How does this finding compare to other insect species compared in this study? i.e., Is this specific to B. dorsalis?

Response: As far as I know, the E78 sequence lacked a DBD in Bactrocera dorsalis and Ceratitis capitate. In contrast, in Drosophila melanogaster and Tribolium castaneum, the sequence has both DBD and LBD.

>L233: type-o in “Because of E75 can accomodate heme...”

Response: Thank you for your advice. We have corrected it.

>In discussions, some of the speculative conclusions could be followed by how to definitely answer the functions of these NRs in B. dorsalis.

Response: Thank you for your advice. We have make some improvement in discussions correspindingly.

>L310: Type-o closed parenthesis: “... related to D. melanogaster HR78) ...”

Response: Thank you for your advice. We have corrected it.

Round 2

Reviewer 1 Report

The authors have done a good job at addressing the highlighted comments. I therefore, only have a few remaining minor comments to further improve the manuscript. All line numbers refer to the corrected manuscript with track changes.

Line 101-102:

In the subsequent blast, the highest result of hits was only one, and there were no multiple results meeting with the requirements. It indicated that no NR duplication events existed in B. dorsalis.

It is not clear to the reviewer exactly what is meant by this statement. Blasting the NR transcripts against B. dorsalis will give many hits with default settings, so it is necessary to show that one hit in particular is much more closely related to the Drosophila gene than all the others. One way to do this is reciprocal blast (performed by the authors), but the wording here makes it sound like taking the top blast hit was enough to exclude the possibility of duplication events. There is no need to redo the analysis, but the wording is still slightly confusing.

Lines 394-395:

The results will contribute to identifying NR genes as insecticide targets and provide insights into new control strategies for fruit flies.

The reviewer still believes this last sentence of the manuscript is overly speculative with regards to insecticide targets. If this is to be included, there should at least be some rational for why NRs may serve as good targets.

Line 275: “Insect” instead of “Insects”

Line 310-313:

In D. melanogaster, DmHNF4 was required in the fat body and insulin-producing cells to maintain glucose homeostasis by supporting a developmental switch toward oxidative phosphorylation and glucose-stimulated insulin secretion at the transition to adulthood

This sentence is far too close to the wording used in Barry and Thummel 2016. While this paper is cited at the end of this sentence, it still represents a case or borderline plagiarism and the wording or content should be changed accordingly.

Author Response

The authors have done a good job at addressing the highlighted comments. I therefore, only have a few remaining minor comments to further improve the manuscript. All line numbers refer to the corrected manuscript with track changes.

Line 101-102:

In the subsequent blast, the highest result of hits was only one, and there were no multiple results meeting with the requirements. It indicated that no NR duplication events existed in B. dorsalis.

It is not clear to the reviewer exactly what is meant by this statement. Blasting the NR transcripts against B. dorsalis will give many hits with default settings, so it is necessary to show that one hit in particular is much more closely related to the Drosophila gene than all the others. One way to do this is reciprocal blast (performed by the authors), but the wording here makes it sound like taking the top blast hit was enough to exclude the possibility of duplication events. There is no need to redo the analysis, but the wording is still slightly confusing.

Response: Thank you. To describe clearly, the description “It indicated that no NR duplication events existed in B. dorsalis” was moved to the end of the paragraph.

Lines 394-395:

The results will contribute to identifying NR genes as insecticide targets and provide insights into new control strategies for fruit flies.

The reviewer still believes this last sentence of the is overly speculative with regards to insecticide targets. If this is to be included, there should at least be some rational for why NRs may serve as good targets.

Response: Thank you for your advice. Because the sentence is overly speculative, the statement was deleted.

Line 275: “Insect” instead of “Insects”

Response: Thanks. Done.

Line 310-313:

In D. melanogaster, DmHNF4 was required in the fat body and insulin-producing cells to maintain glucose homeostasis by supporting a developmental switch toward oxidative phosphorylation and glucose-stimulated insulin secretion at the transition to adulthood

This sentence is far too close to the wording used in Barry and Thummel 2016. While this paper is cited at the end of this sentence, it still represents a case or borderline plagiarism and the wording or content should be changed accordingly.

Response: Thanks. The sentence was changed to “In D. melanogaster, DmHNF4 plays an important role in maintaining glucose homeostasis”.

Reviewer 2 Report

In the current version (v2), the authors have substantially improved the methods section and have added raw data in supplementary tables that helps interpret the results. I would suggest a English grammar review. In  supplementary tables S4, S5 and S6, it is mentioned that "The letters a, b, c, and d after the numbers represent the significance of each gene's different ..." However, I am still not sure what the letters a, b, c and d mean. This should be explained better.

Author Response

In the current version (v2), the authors have substantially improved the methods section and have added raw data in supplementary tables that helps interpret the results. I would suggest a English grammar review. In  supplementary tables S4, S5 and S6, it is mentioned that "The letters a, b, c, and d after the numbers represent the significance of each gene's different ..." However, I am still not sure what the letters a, b, c and d mean. This should be explained better.

Response: Thank you for your advice. We have checked English grammar throughout the manuscript. Besides, in  supplementary tables S4, S5 and S6, different letters after the numbers indicate significant difference at P value < 0.05.